# Management of Endocrine and Metabolic Toxicities of Immune-Checkpoint Inhibitors: From Clinical Studies to a Real-Life Scenario

**DOI:** 10.3390/cancers15010246

**Published:** 2022-12-30

**Authors:** Calogera Claudia Spagnolo, Giuseppe Giuffrida, Salvatore Cannavò, Tindara Franchina, Nicola Silvestris, Rosaria Maddalena Ruggeri, Mariacarmela Santarpia

**Affiliations:** 1Medical Oncology Unit, Department of Human Pathology “G.Barresi”, University of Messina, 98125 Messina, Italy; 2Endocrinology Unit, Department of Human Pathology of Adulthood and Childhood DETEV, University of Messina, 98125 Messina, Italy

**Keywords:** immunotherapy, immune checkpoint inhibitors (ICIs), immune-related toxicity, endocrine-related toxicity, predictive biomarkers, multidisciplinary treatment

## Abstract

**Simple Summary:**

Immune checkpoint inhibitors currently represent the standard of care for the treatment of different tumor types and have also been proven to be effective in several disease settings. However, their use is associated with a peculiar toxicity profile, related to the enhancement of the immune response, affecting several organs. The identification of predictive biomarkers has a crucial importance to select those patients that can better benefit from immunotherapy, improving their outcomes, while potentially avoiding toxicities with these drugs. In this review we will include the most recent data and current knowledge on immune-related endocrine and metabolic adverse events and on biomarkers and risk factors with a notable predictive value for their incidence. Furthermore, we will summarize the latest studies and recommendations on the clinical approach to these types of adverse events with the purpose of optimizing the diagnostic algorithm and their therapeutic management.

**Abstract:**

Immune checkpoint inhibitors (ICIs) have revolutionized the therapeutic landscape of solid tumors. However, although ICIs are better tolerated than conventional chemotherapy, their use is associated with a peculiar toxicity profile, related to the enhancement of the immune response, affecting several organs. Among immune-related adverse events (irAEs), up to 10% involve the endocrine system. Most of them are represented by thyroid disorders (hypothyroidism and hyperthyroidism), mainly correlated to the use of anti-PD-1 and/or anti-PD-L1 agents. Less common endocrine irAEs include hypophysitis, adrenalitis, and metabolic irAEs. A deeper understanding of endocrine toxicities is a critical goal for both oncologists and endocrinologists. A strict collaboration between these specialists is mandatory for early recognition and proper treatment of these patients. In this review we will provide a comprehensive overview of endocrine and metabolic adverse events of ICIs, with particular interest in the pathogenesis, predisposing factors and clinical presentation of these irAEs, and their impact on clinical outcomes of patients. Furthermore, we will summarize the most recent studies and recommendations on the clinical approach to immune-related endocrinopathies with the purpose to optimize the diagnostic algorithm, and to help both oncologists and endocrinologists to improve the therapeutic management of these unique types of irAEs, in a real-life scenario.

## 1. Introduction

In the last decade, cancer immunotherapy has revolutionized the therapeutic landscape of solid tumors, restoring and improving the host’s natural immune response against cancer cells [1]. Immune checkpoints, such as cytotoxic T-cell antigen 4 (CTLA-4) and programmed death 1 (PD-1) and its ligand PD-L1, act as negative co-regulators, maintaining self-tolerance and regulating duration and dimension of T lymphocyte responses, thus reducing damage to normal cells, following an excessively prolonged activity of the immune response, as in the case of infections. The CTLA-4 is induced in T-cells at the time of their initial activation and response to an antigen, whereas the major role of the PD-1/PD-L1 pathway is to regulate inflammatory responses by effector T-cells, upon recognizing antigens in peripheral tissues [2].

The expression of PD-L1 on solid tumors and the key role of the PD-1/PD-L1 and CTLA-4 pathways in tumors, as mechanisms to evade the host’s immune system, represent the biological rationale for their use as potential targets in cancer treatment.

Immune checkpoint inhibitors (ICIs) are monoclonal antibodies that inhibit immune checkpoints, improving the response of immune cells and restoring the antitumor activity of cytotoxic T-lymphocytes [2]. Currently, various agents have been approved by the US Food and Drug Administration (FDA) for several tumors; these include anti-PD-1 (nivolumab, pembrolizumab, and cemiplimab), anti-PD-L1 (atezolizumab, avelumab, durvalumab, and dostarlimab) antibodies, anti-CTLA-4 (ipilimumab and tremelimumab), and anti-lymphocyte-activation gene 3 (LAG-3) (relatlimab) antibodies, used as single-agents or in combinatorial regimens [3,4].

These drugs are generally better tolerated, compared to conventional chemotherapy. However, the increase in immune response that they produce can be responsible for a peculiar toxicity profile that has been widely described and characterized. The incidence of toxicity is estimated in the range of 54% to 76% and is more frequently associated with the use of combination regimens or anti-CTLA-4 drugs, compared to both anti-PD-1 and anti-PD-L1 drugs alone [5].

Several immune related adverse events (IrAEs) are correlated with the use of anti-CTLA-4, such as skin reactions (44%) and gastrointestinal (GI) disorders including colitis (35%), whereas irAEs associated with the use of anti-PD-1 and/or anti-PD-L1 are mainly represented by endocrinopathies (5–20%) and pneumonia (2–5%) [6,7,8]. Hypothyroidism (8–10%) and hyperthyroidism (6%) are the most frequent endocrine toxicities related to ICI treatment, mainly to anti-PD-1/PD-L1 agents. Less common endocrine irAEs are hypophysitis, adrenalitis, and metabolic irAEs, represented by an autoimmune type 1 diabetes mellitus (DM) [9,10].

In this review we provide a comprehensive overview of endocrine and metabolic adverse events of ICIs, with particular regard to the pathogenesis, predisposing factors, and clinical presentation of irAEs, and their impact on clinical outcome. The purpose is to optimize the diagnostic algorithm and help clinicians to improve the therapeutic management of these unique types of irAEs in a real-life scenario.

## 2. Immune Checkpoints Role and Rationale for ICIs Use

In humans, the immune system is responsible for recognizing and eliminating pathogens, including cancer cells, by mediating non-self antigen detection. Innate immunity, characterized by a fast and non-specific response, coexists with adaptive immunity, that acts with a delayed but specific response, and is an expression of immunological memory [11]. Cytotoxic T-lymphocytes (CTLs) represent the main immunity effectors, but other cell populations like CD4+ lymphocytes, CD8+ lymphocytes, natural killer cells (NK), natural killer T-cells (NKT), and B-lymphocytes are also involved [12,13]. An effective cytotoxic attack requires an appropriate antigen presentation by antigen presenting cells (APC), mainly macrophages and dendritic cells (DCs) [14].

Tumor cells can escape immune surveillance by several mechanisms. According to the spatial distribution of immune cells in the tumor microenvironment (TME), a tumor can be classified into one of three immunophenotypes: immune-inflamed, immune- excluded, and immune-desert phenotype. The immune-inflamed phenotype is characterized by high tumor- infiltrating lymphocytes (TILs), increased interferon-γ (IFN-γ) signalling, high expression of PD-L1, high tumor mutational burden (TMB), and are also called “hot tumors”. The immune-excluded and immune-desert phenotypes are characterized by rare tumor- infiltrating lymphocytes (TILs), low TMB, low major histocompatibility complex (MHC) class I, weak PD-L1 expression levels, and they are defined as “cold tumors”. In contrast to the inflamed phenotype, they are characterized by an ineffective immune response and for this reason they hardly respond to ICI therapy [15,16,17].

In tumors, CTLA-4 and PD-1/PD-L1 can favor tumor evasion from immune surveillance, so that their inhibition can produce an increased immune activation and overcome the typical tumor-induced immunosuppression [18,19].

CTLA-4 is a molecule widely expressed on regulatory T-cells (Tregs), which facilitates their regulatory activity and competes with the CD28 co-stimulatory receptor for the binding to CD80 and CD86 ligands, expressed on APCs. Its increased affinity for these ligands produces an inhibitory signal that causes the depletion of T-cells antitumoral activities and proliferative block, with the consequent survival of neoplastic cells. It is responsible for early regulation of T-cell proliferation, at the lymph node level [20,21,22]. PD-1 is an inhibitory transmembrane receptor of the immunoglobulin superfamily B7, localized on surface of T helper, T cytotoxic, T regulatory cells, B lymphocytes, and NK cells. The expression of PD-1 on T cells plays an important role within the context of a cytotoxic effect inhibition, since its interaction with ligands on tumor cells results in lymphocyte apoptosis, by Fas receptor induction [23]. PD-L1 (B7-H1, CD274) is expressed on T and B cells, macrophages, and DCs, but also on nonlymphoid cells (myeloid cells, endothelial cells, cardiac cells, and muscle cells) [24]. Increased PD-L1 and PD-L2 (CD273, B7-DC) levels have also been found on several tumor types, suggesting a prominent role of PD-1/PD-L1 axis as a mechanism to escape immune surveillance, with a later action in peripheral tissues, and tumor microenvironment (TME) [20]. Another mechanism that hides cancer cells from CTL attack is represented by a costimulatory molecule expressed both on cancer cells and APCs, B7. The interaction of B7 molecules (CD80/CD86) on the APC with the CD28 receptor on lymphocyte, is mandatory to induce the cytotoxic effect. B7 molecules may also mediate an inhibitory effect through the association with CTLA-4 [25,26].

The strategy of inhibiting the above pathways allows the unmasking of the immune response, which becomes paralyzed by the inhibitory mechanisms, caused by the tumor, and to restore the antitumor activity of cytotoxic T-lymphocytes. Therefore, ICIs are monoclonal antibodies that work by blocking the binding of checkpoint proteins to partner proteins, preventing the inhibition signals, and allowing T-cells to kill cancer cells [2]. On the other hand, checkpoint blockade can also lead to a reduced self-tolerance against other tissues, thus favoring autoimmune processes [27] (Figure 1).

Despite their different mechanisms of action, ICIs present a similar molecular structure. They are monoclonal antibodies (mAbs), with a molecular weight of about 150 kDa, which share several pharmacokinetic properties [28,29]. Anti-CTLA-4 and anti-PD-L1 are immunoglobulin (Ig) G1, while anti-PD-1 are IgG4. IgG1 mAbs have a higher capacity for activating antibody-dependent cellular cytotoxicity (ADCC) and complement-dependent cytotoxicity (CDC) than IgG4 [30,31]. Since mAbs are unable to permeate through the intestine (due to polarity and size) [32,33], ICIs are administered by intravenous infusion, achieving complete bioavailability within hours of injections. Furthermore, the distribution of mAbs to target cells is aided by the lymphatic system, and influenced by the rate of lymphatic flow. Therefore, due to their pharmacokinetics properties, these drugs, could also be administered subcutaneously [29].

Since 2011, when the anti-CTLA-4 Ipilimumab first obtained FDA approval for the treatment of metastatic melanoma [34,35], the number of ICIs has rapidly expanded to include indications in different tumor types and disease settings (Table 1) (www.fda.gov, accessed on 15 November 2022).

Recently, an anti-LAG 3 antibody (relatlimab), has been approved for the treatment of metastatic melanoma in combination with nivolumab. Similar to other checkpoint proteins, LAG-3 is a molecule expressed on the surface of effector and regulatory T-cells (Tregs) that modulate T-cell activation, mediating a decreased capacity to attack tumor cells. Relatlimab, an inhibitory antibody targeting LAG-3 on T-cells, can restore the effector function of T-cells [36]. In addition to the ICIs already approved by the FDA, several antibodies and small molecules targeting other immune checkpoints such as TIM3, TIGIT, BTLA, and agonists of the co-stimulatory receptors GITR, OX40, 41BB, and ICOS, are in clinical development [37].

## 3. Physiopathology of ICI-Related Adverse Effects

Although ICIs are generally better tolerated than conventional chemotherapy modalities, they present a unique toxicity profile directly originating from their mechanism of action. In fact, irAEs are typical autoimmune reactions whose overall incidence ranges from 15% to 90% [38,39]. According to evidence, their entity is proportional to an adequate immune response towards tumor cells, so that long responders can also bear the greatest risk for (chronic) irAEs. In fact, shared T-cell receptor sequences and up-regulated organ-specific transcripts have been demonstrated both in tumors and in normal tissues affected by toxicities [40]. IrAEs can potentially involve any district in the body, mainly interesting endocrine and gastrointestinal systems, skin, and liver. Inflammatory events, such as cardiovascular, hematologic, renal, neurologic, and ophthalmic, are less common [38]. However, this variegated spectrum can also be the expression of pathophysiological mechanisms not related to the anti-tumor function, but to other factors, including the microbiome, viral factors, or tissue-specific factors. For example, it has been reported that tissue-resident memory CD8+ T-cells were found as predominant immune cells in colon biopsy samples obtained from a cohort of patients experiencing ICI-induced colitis [40].

The US National Cancer Institute has developed a score for irAEs, according to the Common Terminology Criteria for Adverse Events (CTCAE), that includes a toxicity rank varying from moderate (1–2) to severe (3–4), up to death (5) [41]. In fact, although most irAEs are low-grade reactions, and are easily manageable when adequately treated, they can become extremely severe, up to life-threatening, if incorrectly managed or left untreated, leading to the patient’s *exitus*, in some cases.

## 4. Endocrine Adverse Effects

In general, almost 80–90% of subjects undergoing ICI therapy can develop any type of irAE, with severe adverse effects occurring in 15–20% of patients treated with anti-PD-1/PD-L1, and over 60% under combined treatment with anti-CTLA-4 *plus* anti-PD-1/PD-L1 [42]. Interestingly, only anti-CTLA-4 agents cause irAEs in a dose-dependent fashion, while pre-existing autoimmune disorders or organ dysfunctions (i.e., NAFLD, smoking habits) seem to increase the risk. Regarding endocrine irAEs, it is worth highlighting that these adverse effects, which usually lead to chronic conditions and require lifelong therapies, differ from other events, mostly linked to acute inflammatory presentations, which are often transient and well controlled with steroid administration. This could be explained with the destruction of a relatively low number of hormone-secreting cells [42].

Endocrine irAEs of any grade occur from 10% up to 40% of patients under ICIs, with variable rates of endocrine dysfunction (0–40%) in different studies [43]. IrAE clinical presentation and management do not differ according to the type of drug used, and among them, the most reported are—from the most frequent to rarer forms—thyroid dysfunction (usually hypothyroidism, sometimes preceded by transient thyrotoxicosis, or less frequently autoimmune hyperthyroidism from Graves’ disease—GD), hypophysitis, adrenal insufficiency, and autoimmune DM from pancreatic islet cell insufficiency [44,45,46]. The average presentation of endocrine dysfunctions is within 6 months from the start of treatment, with a median onset of 9–11 weeks after the first dose (range 5–36 weeks), but later toxicities have also been reported [43,45]. The combination of anti-CTLA-4 and anti-PD-1/PD-L1 has been associated with early endocrine irAEs, often with multiple manifestations [44,45,46,47,48].

### 4.1. Thyroid Disorders

#### 4.1.1. Epidemiology and Clinical Presentation

Thyroid disorders are the most frequent endocrine adverse reactions, being more often related to anti-PD-1 therapy (up to 40%), and are almost the only endocrine side effect of anti-PD-L1 (6–11%), compared to anti-CTLA-4 (5% of cases) [10,48]. Their incidence also increases (up to 15–20%) under combination therapy [27,42,49]. The majority of ICI-related thyroid toxicities are mild or moderate (grade 1 or 2, according to CTCAE), while severe complications (grade 3 or 4) are less frequent, and mostly correlated to combination regimens [50]. Thyroid-related AEs more frequently involve women and range from overt hypothyroidism to hyperthyroidism (including Graves’ orbitopathy). Rarely, severe forms including thyroid storm, myxedema, or steroid-responsive encephalopathy have been observed [43,45,46]. Hypothyroidism is the most frequent presentation (3.9–8.5%), preceded by destructive thyroiditis in 30–40% of patients treated with anti-PD-1/anti-PD-L1, and two thirds of patients under combined treatment [27,42]. Destructive (and often painless) thyroiditis, as an autoimmune-mediated inflammatory disease, presents with the classic triphasic course: (a) transient thyrotoxicosis due to thyroid hormone release; (b) subclinical or overt hypothyroidism; and (c) subsequent recovery to normal thyroid function [10,51,52,53]. However, in several patients undergoing ICI treatment, hypothyroidism duration can be prolonged until it becomes permanent. Chronic autoimmune thyroiditis (Hashimoto’s thyroiditis), characterized by anti- thyroid peroxidase (TPO-Ab) and anti-thyroglobulin (Tg-Ab) antibodies in more than one-third of patients can occur, and present with either subclinical or overt hypothyroidism [47,48,51,52,53]. It is still unknown if their presence boosts thyroid dysfunction, or they originate from an immunological response against the antigen release, following a destructive thyroiditis [27]. On the other hand, pre-existing anti-thyroid antibodies and their titer seem to increase the risk of thyroid dysfunction (20–50% vs. 1–2.5% in antibody-negative patients) due to a further removal of self-tolerance, that on the converse could be correlated to improved overall survival (OS), despite the mechanism for this correlation not yet being clear [42,54,55,56,57,58,59,60,61,62].

These findings have been confirmed in another study by Rubino et al., in which the authors evaluated the development of endocrine irAEs in a cohort of patients treated with anti-PD-1. Out of 251 patients, 70 (27.89%) presented with endocrine irAEs, being for the most part thyroid dysfunctions (94%), mainly hypothyroidism. Female sex and pre-existing thyroid autoimmunity were significantly linked to endocrine irAE development, generally within 6 months from treatment initiation. Moreover, the appearance of any autoimmune toxicity, even of a non-endocrine nature, was related to a better outcome in terms of progression-free survival (PFS) and overall survival (OS) [63]. An overview of the incidence of thyroid irAEs is shown in Table 2.

Additionally, significant thyroid uptake at ^18^FDG scintigraphy before ICI treatment initiation could be predictive of thyroid alterations, likely because such a diffuse glandular uptake is usually related to a pre-existing chronic thyroiditis. Moreover, histologic evaluation of samples from subjects with autoimmune thyroiditis (both Hashimoto’s and painless/destructive thyroiditis) demonstrated lymphocytic infiltration by B-cells and cytotoxic T-cells. PD-1 is expressed by these cells (and also NK cells), so that its inhibition causes their proliferation and more frequent thyroid dysfunctions than anti-CTLA-4, which only stimulate T-lymphocytes [27].

Thyrotoxicosis is less common (1–6.5% of patients), with differences according to drug class having low frequencies with ipilimumab (0.2–1.7%), higher frequency with anti-PD-1/anti-PD-L1 (0.6–3.7%), and the highest frequencies during combination therapy (8.0–11.1%) [10,50]. Furthermore, the risk for thyrotoxicosis is significantly higher with anti-PD-1 compared to anti-PD-L1 antibodies, even with same-class differences (pembrolizumab higher than nivolumab). As stated before, thyrotoxicosis is more often related to a transient destructive thyroiditis, while fewer cases are due to Graves’ disease (GD) [47,48,51,52,53]. Several cases of Graves’ ophthalmopathy (GO) have also been reported [27].

#### 4.1.2. Diagnosis and Treatment

Hypothyroidism symptoms can be very mild at onset, or they might be ascribed to the underlying disease and/or concomitant multiple medications taken by these patients. A clinical picture characterized by increasing fatigue, weight gain, cold intolerance, constipation, and depression should raise a significant suspicion for thyroid dysfunction. Biochemically, thyroid stimulating hormone (TSH) elevation is the earliest marker of thyroid dysfunction, and it can range from 4 to 10 μUI/mL with normal FT3 and FT4 in subclinical presentations, while it can be elevated (over 10 μUI/mL) with reduced free hormones, in overt hypothyroidism. Furthermore, moderate hypercholesterolemia, and, although nonspecific, mild anemia can be observed in these settings. Mild forms of subclinical hypothyroidism with slight TSH elevations and no symptoms can be monitored with consecutive blood tests, concomitant to treatment administration. In the presence of clinically evident hypothyroidism with mild/moderate TSH elevations, the mainstay of treatment is levothyroxine, that can be initially administered at low doses (25–50 mcg/d) and then up titrated, if necessary, with 4–6-week intervals. Lower doses (12.5–25 mcg/d) should be used early or in people with coexistent cardiovascular diseases. As it is well known, primary/central adrenal insufficiency should be ruled out—if suspected—and treated before starting levothyroxine, or this can precipitate an adrenal crisis, with life-threatening consequences [10,27]. The newer liquid formulations/soft gel caps should be considered. In fact, as demonstrated in cases of severe malabsorption, compared to traditional levothyroxine tablets, that require fasting and at least a 30 min-interval from breakfast, their pharmacokinetics could ameliorate both patient adherence and dose absorption in the context of multidrug regimens [64].

On the converse, thyrotoxicosis presents with signs and symptoms of thyroid hormone excess, like weight loss, heat intolerance, sweating, palpitations, tremors, and diarrhea, but such manifestations are usually less evident than in GD, or the patient can also be asymptomatic. However, hyperthyroidism symptom expression can be reduced by concomitant medications (i.e., beta-blockers) or advanced age (“apathetic” forms in the elderly) [65,66]. Biochemically, reduced TSH levels (<0.4 μUI/mL) are paired with normal (in subclinical forms) or elevated free triiodothyronine (FT3) and FT4. GD should be suspected with an increased serum FT3/FT4 ratio, with a suggestive ultrasonography pattern (goiter with hypoechogenic and inhomogeneous pattern and increased vascularization), with persistent thyrotoxicosis or concomitant thyroid eye disease. In these cases, TSH receptor antibodies (TRAb) are expected to be elevated. On the other hand, in thyrotoxicosis from destructive/painless thyroiditis, a reduced FT3/FT4 ratio with negative TRAb, despite the presence of TPO-Ab and/or Tg-Ab, is usually observed. In cases of difficult differential diagnosis, a scintigraphy for radioiodine uptake pattern evaluation can be performed (diffuse and increased in GD/reduced or absent in thyroiditis), although the frequent use of iodine contrast for imaging in these patients can alter proper gland uptake [10,51,52,53,67].

As stated above, cases of GO have also been described, with typical signs and symptoms such as proptosis, eye pain, conjunctival redness, periorbital oedema, ophthalmoplegia, and swelling of extraocular muscles on orbital imaging. It should be noted, however, that there are also some reports on an inflammatory condition, defined thyroid eye disease (TED)-like orbital inflammatory syndrome or inflammatory orbitopathy, characterized by euthyroidism and negative TRAb, independently from increased TPOAb and TgAb levels [68,69,70].

### 4.2. Pituitary Disorders

#### 4.2.1. Epidemiology and Clinical Presentation

Hypophysitis is a typical endocrine side effect of anti-CTLA-4 agents, since as hypothesized in some studies, CTLA-4 is expressed in prolactin (PRL) and TSH-secreting pituitary cells. In this context, ipilimumab and tremelimumab (via IgG1 and IgG2, respectively) activate complement and start a type II hypersensitivity reaction with antibody-dependent cell-mediated cytotoxicity. Other evidence shows the appearance of anti-pituitary antibodies (APA) anti-TSH, FSH-, and adrenocorticotropic hormone (ACTH) secreting cells in patients treated with anti-CTLA-4 [71,72]. On the contrary, hypophysitis is less frequent with anti-PD-1/PD-L1, that cannot activate the complement cascade via IgG4, but are also expressed on pituitary cells, thus leading to a sort of IgG4-mediated hypophysitis [72]. Hence, hypophysitis has an incidence of 0–17.4% for ipilimumab and 0.4–5% for tremelimumab, with a dose-dependent relationship [43,45,46,47,48]. It is an uncommon irAE of anti-PD-1 drugs (0.3–1.1%), and it has never been reported with anti-PD-L1, while combination therapy significantly increases the risk from 7.7% to 10% [27,72]. There are no other specific risk factors, but some authors have reported higher incidence of hypophysitis in male patients of older age, with a possible explanation lying in the positive effect of androgens, on CTLA-4 expression [72,73,74]. Median onset is at about 11 weeks, usually within 6–12 weeks after treatment initiation [45]. In most cases, it is restricted to the anterior pituitary (adenohypophysitis), with usually evident clinical symptoms and signs due to either hormonal deficiency (from isolated deficits up to panhypopituitarism) or mass effect (headache, asthenia, nausea, weakness and anorexia, hypotension, oligo-amenorrhea in females, erectile dysfunction in males, and loss of libido) [72,75,76]. However, isolated hormone deficits can be more frequent, thus the clinical presentation of ICI-related hypophysitis significantly differs from classic lymphocytic autoimmune hypophysitis (LAH). In fact, ACTH and TSH secretion are frequently compromised, while GH deficit is the rarest. Gonadal axis is involved in up to 83–87% of male patients, while prolactin can be elevated or low in these patients. Conversely, symptoms at presentation may sometimes be nonspecific, especially fatigue, which is the most common [72,75,76]. Indeed, some recent studies demonstrated that patients with circulating APA at baseline were more prone to develop an isolated ACTH deficiency after ICI administration, characterized by fatigue and hyponatremia as typical hallmarks. This finding also seemed to be related to specific HLA loci (HLA-Cw12, HLADR15, HLA-DQ7, and HLA-DPw9), and could be more frequent in patients treated with anti-PD-1/PD-L1 [77,78]. Diabetes insipidus is typical of infundibulo-neurohypophysitis, a rare condition involving both the infundibular stem and the posterior lobe. It can less frequently occur in adenohypophysitis, due to an inhibited axonal anti-diuretic hormone (ADH) transport through the infundibulum, and it is characterized by polyuria, polydipsia and/or hypernatremia. Of note, in the context of ICI-related hypophysitis, diabetes insipidus is an infrequent condition, generating issues of differential diagnosis with potential pituitary metastases often involving the posterior lobe: in these cases, radiological evaluation by magnetic resonance imaging (MRI) with gadolinium contrast is required (see following paragraph).

#### 4.2.2. Diagnosis and Treatment

One of the proposed diagnostic classifications suggests the presence of ≥1 pituitary deficit (ACTH or TSH) and MRI alterations, or ≥2 pituitary deficits with headache and other symptoms [38]. When hypophysitis is suspected, brain/pituitary MRI should be performed as the gold standard imaging evaluation. During the acute phase, some typical findings can be observed such as gland enlargement, stalk thickening, and homogeneous or heterogenous contrast enhancement, especially in patients under anti-CTLA-4 [72]. Given the frequently rapid onset of symptoms, MRI is also useful to rule out symptomatic brain or pituitary metastases, abscesses, or pituitary apoplexy [79]. However, pituitary metastases often develop in the posterior lobe, more frequently cause diabetes insipidus, and sometimes present with typical MRI features (i.e., “dumbbell” shape); moreover, in some recent reports in the literature, at least part of them seem to incidentally respond to ICI treatment for the primary tumor [79,80]. As from clinical practice, monitoring of ACTH, cortisol, sodium, and potassium levels would be advisable at every cycle of anti-CTLA-4 or combined treatment, while in case of anti-PD-1/PD-L1 therapy, checking adrenal function is not mandatory, but it depends on the presence of symp-toms suggesting a potential condition of hypoadrenalism [10,72]. In the presence of dehydration, hypotension or shock symptoms an adrenal crisis should be suspected: it is a life-threatening condition that requires prompt corticosteroid administration, even if it more frequently occurs in cases of primary adrenal insufficiency (PAI)—rarer under ICIs. On the converse, frankly reduced cortisol levels (≤5 μg/dL) are diagnostic of secondary hypoadrenalism, as per guidelines, while in cases of borderline alterations, even a low-dose cosyntropin test can be performed (diagnosis confirmed if cortisol levels do not raise over 18–20 μg/dL). Treatment is provided with the administration of corticosteroids (i.e., hydrocortisone starting from 10–20 mg/d in fractioned doses) before any other replacement therapy. Then, thyroid hormone can be introduced (remember that thyroid replacement in the context of untreated adrenal insufficiency can cause an adrenal crisis), and further treatment with testosterone/estrogen can be considered if indicated. Patients with severe compressive symptoms can receive prednisone 1–2 mg/kg with a rapid taper (over 1–2 weeks). Diabetes insipidus can be managed with oral/nasal desmopressin administration, as per current guidelines [10,42,72].

### 4.3. Adrenal Disorders

#### 4.3.1. Epidemiology and Clinical Presentation

Primary adrenal insufficiency (PAI) is a less common endocrine irAE of ICIs, and it can be due to adrenalitis, an autoimmune inflammation of the adrenal glands. Although overall incidence is low, it occurs more frequently with the use of PD-1 antibodies (0–4.3% with pembrolizumab and 0–3.3% with nivolumab) than with CTLA-4 inhibitors (ipilimumab 0.3–1.5%) [43,45,81]. Among the few cases reported, most patients were male, with a mean age of 52 years and a median onset of 10 weeks, after ICI treatment initiation [27]. It is generally characterized by a double hormone deficiency involving both glucocorticoids (cortisol) and mineralocorticoids (aldosterone). For this reason, PAI presentation—especially when the inflammatory process is at an advanced stage—is usually more severe than secondary forms. It can present with nausea/vomiting, fatigue, postural or systolic hypotension, tachycardia, anorexia, abdominal pain, and weight loss. Hyponatremia and/or hyperkalemia are markers of mineralocorticoid deficiency, usually associated with low plasma volume. However, regarding hyponatremia in these patients, it can be caused by other conditions that need to be investigated and differentiated, like syndrome of inappropriate secretion of anti-diuretic hormone (SIADH), due to ectopic ADH secretion from malignancy, other drugs, acute illnesses, etc. Finally, in the most severe form of PAI, adrenal crisis, clinical picture is characterized by hypovolemic shock with fever, generalized abdominal pain, and vomiting up to confusion/coma [10].

#### 4.3.2. Diagnosis and Treatment

Regular monitoring of electrolytes (and frequently, morning cortisol) is advisable in patients treated with ICIs. For PAI diagnosis, morning blood samples should be obtained for measuring cortisol, ACTH, renin, and aldosterone. Reduced cortisol and aldosterone with elevated ACTH and renin levels confirm both glucocorticoid and mineralocorticoid deficiency. In cases of inconclusive cortisol values, as already stated for hypophysitis, a cosyntropin test should be performed [10,27]. Especially when issues with differential diagnosis occur (i.e., suspecting adrenal metastases), computed tomography (CT) imaging of the abdomen can be performed, showing bilateral adrenal enlargement. The same finding in follow-up CTs, in patients treated with ICIs, should orient towards benignity [82]. As in the case of central hypoadrenalism, treatment is provided by use of glucocorticoids (i.e., hydrocortisone in fractionated daily doses). In the presence of adrenal crisis, or when an acute adrenal insufficiency is suspected, stress-dose glucocorticoids (GCS) should be immediately administered. A typical regimen includes an initial dose of 100 mg of hydrocortisone iv, followed by 50 mg every 6 h, then gradually tapered to basal replacement dose (15–25 mg/d). Also, long-term fludrocortisone replacement is required in patients with confirmed aldosterone deficiency [27].

## 5. Metabolic Adverse Effects

### 5.1. Diabetes

#### 5.1.1. Epidemiology and Clinical Presentation

Type 1, or autoimmune diabetes mellitus (T1DM) is characterized by pancreatic B-cell destruction mediated by autoreactive T-cells. The main factor is the self-immunity of B-lymphocytes, which is mainly found in patients with HLA-DR3-DQ2 or HLADR4-DQ8 haplotypes or both, whereas environmental factors influence the development of this disease [83,84]. PD-L1 is expressed in pancreatic islet cells, and the PD-1/PD-L1 interaction has been shown to play a protective role for the onset of T1DM, by inhibiting the activation of autoreactive T-lymphocytes [85]. A report evaluating pancreatic tissue from a patient with ICI-associated diabetes mellitus (ICI-DM), showed an enhanced peri-islet CD8+ T-cell infiltration and, in contrast to classic T1DM, complete depletion of insulin-positive cells, suggesting that ICI-DM may be correlated to a more rapid and severe necrosis/apoptosis of pancreatic beta cells than T1D [86]. About 97% of all reported cases of ICI-DM have arisen with anti-PD-1/PD-L1 or combination regimens, whereas reports of cases on CTLA-4 monotherapy are rare [87,88,89]. Time of onset is variable and ranges between 1 and 228 weeks [90]. In most cases, it presents as fulminant diabetes associated with diabetic ketoacidosis, suggesting acuity and rapid onset, with a sudden decline in insulin secretory function. In a study of 27 patients, ketoacidosis development was found in 57% of cases, while 42% of cases were characterized by pancreatitis, with a mean HbA1c levels of 7.95% and decreased C peptide levels [91]. Other presenting symptoms include polyuria, polydipsia, fatigue, abdominal pain, and nausea [42].

#### 5.1.2. Diagnosis and Treatment

The primary monitoring strategy is random blood glucose measurements at each ICI cycle. In patients with new-onset hyperglycemia, peptide C and A1C should be measured, which may help distinguish the etiology of hyperglycemia (steroid-induced hyperglycemia, type 2 diabetes, stress-induced, etc.) [42].

Tests for insulin antibodies (anti-insulin, anti-insulin-cell A, glutamic acid decarboxylase, and zinc transporter 8), C-peptide, and insulin could be used to distinguish between T1DM and T2DM.

The treatment of choice is represented by insulin injections aimed at maintaining an HbA1c lower than 8.0%. As with other endocrinopathies, corticosteroid therapy is not indicated. No evidence suggest that steroids can help with gland function [7,92].

## 6. Potential Biomarkers of irAEs Development and irAEs Impact on Clinical Outcomes

More than a decade after ICIs were first studied in clinical trials, there are still no clear predictors of irAEs in treated patients, but several hypotheses have been provided in the literature. As stated above, male sex can be a risk factor for the development of autoimmune hypophysitis [72,73,74]. Moreover, recent studies have identified some autoantibodies expressed in human pituitary gland and associated with autoimmune hypophysitis: anti-guanine nucleotide-binding protein G subunit alpha (GNAL), anti-integral membrane protein 2B (ITM2B), and anti-ZCCHC8. These molecules can be considered as potential predictive biomarkers for the onset of hypophysitis [93]. As above reported, female sex and previous thyroid autoimmune disease can be correlated to thyroid irAEs onset, as well as a significant thyroid uptake at ^18^FDG scintigraphy, before treatment initiation [27,63]. Furthermore, several studies reported that the presence of both anti-thyroid peroxidase antibody at baseline and anti-thyroglobulin antibody during the therapy was significantly correlated to hypothyroidism development [59,94,95]. Stamatouli et al., have shown that the HLA typing is correlated with the development of ICI-induced autoimmune insulin dependent diabetes, and in particular, HLA-DR4, most frequently (76%) associated with the disease [91]. Furthermore, other studies have shown that both increased eosinophils and basophils levels are correlated to enhanced onset of irAEs involved mainly in the endocrine system and skin, compared to other organs [96,97].

On the other hand, the research to identify predictive biomarkers of ICI response is still ongoing. Up to now, it has mainly been focused on tumor signatures (PD-L1 expression, MSI-H status, and tumor mutational burden—TMB) [98,99,100,101]. Other exploratory biomarkers include DNA repair gene mutations, immune cell exclusion, oncogenic signaling pathways, HLA genotype, baseline levels of innate immune cells, circulating blood cell counts, microRNA, cytokines, autoantibodies, and serum proteins [102,103]. In addition to these tissue- and blood-based parameters, some clinical characteristics have been associated with ICI efficacy (e.g., exposure to steroids, microbiome, smoking, and weight) [104,105,106]. One of the most impressive observations is the association between ICI benefit and irAE onset, as an expression of autoimmune toxicities. Several studies have demonstrated an association across ICI types, with some evidence showing how both increased severity and number of irAEs can be positively correlated to more pronounced clinical benefit with these agents. Particularly, in patients developing irAEs under anti-PD-1/PD-L1 therapy, better outcomes in terms of overall response rate (ORR), PFS, and OS have been reported [107,108,109,110,111,112], while in patients treated with anti-CTLA-4 antibodies, this association was less consistent [113,114,115,116]. In a meta-analysis by Hussaini et al., the correlation between the incidence of irAEs after use of ICIs and clinical outcomes has been assessed. This work included data from 51 studies assessing ICIs in different types of advanced cancer (melanoma, lung, renal, urothelial, head and neck, and gastrointestinal tumors). A positive, independent correlation was found between irAE onset and PFS, OS, and ORR, in patients treated with ICIs, regardless of the disease site, type of ICI, and irAE. Grade 3 or more toxicities correlated with better ORR, but worse OS [117]. A recent study analyzing two independent cohorts including a large number of NSCLC patients treated with PD-1/PD-L1 inhibitors, found that patients developing irAEs were characterized by more favorable outcomes. Interestingly, this study has also shown that late-onset (>3 months) irAE development was correlated to this clinical benefit, while patients with earlier toxicities showed statistically and clinically significant worse outcomes [103].

Specifically focusing on endocrine irAEs, a study by Paderi et al., evaluated the incidence of irAEs in 43 patients with metastatic renal cell carcinoma treated with nivolumab or nivolumab plus ipilimumab. In the studied cohort, 49 different irAEs were observed in 29 patients (67.4%), with thyroid dysfunction the most frequent irAE (14 out of 29). Interestingly, patients with thyroid dysfunction and cutaneous reactions presented with a significantly longer median PFS after treatment start, and thyroid dysfunction was an independent predictor of favorable outcome. Furthermore, the occurrence of ≥2 irAEs in the same patient correlated with better outcomes, but a clear pathophysiological explanation for this finding is still lacking [54]. Another study by Karhapaa et al. evaluated the relationship between irAEs and outcomes in 140 patients affected by metastatic melanoma and treated with ipilimumab (15%), nivolumab (33%), pembrolizumab (48%), and combination therapy (6, 4%). Among the studied subjects, endocrine irAEs were recorded in 41 (29%) patients and thyroid disorders were the most frequent manifestations occurring in 26%. Even in this case, significantly better outcomes were reported in those who experienced endocrine irAEs, with a median PFS of 8.1 versus 2.7 months, and a median OS of 47.5 versus 23.7 months [55]. Furthermore, in a retrospective observational study that included 570 patients with different advanced tumor types receiving treatment with ICIs, documented a significantly better survival in patients that develop endocrine-related AEs, compared to patients without (*p* < 0.001) [118]. Due to the frequency of thyroid irAEs, they have been largely studied and associated with improvement in PFS and OS [56,57,58,59]. Growing data show that thyroid irAE subtypes are not equally correlated with clinical outcomes. When thyroid irAEs are biochemically evident, there is a significant correlation with improvement in terms of PFS and OS, when compared to the subclinical ones [60,61,62]. Moreover, irreversible thyroid dysfunction with initiation of replacement therapy, has been associated with improved outcomes [119,120,121].

However, despite the above observations, future prospective studies are needed to identify biomarkers and risk factors with notable predictive values, for the incidence of irAEs induced by ICIs, and to clarify the temporal relationship between anti-tumor and anti-host effects of ICIs, and further demonstrate the association of different irAEs with clinical outcomes.

## 7. Multidisciplinary Management of irAEs in Real-Life

The frequency of endocrine irAEs with long-term persistence during ICIs treatment is one of the aspects that makes the need for a multidisciplinary approach mandatory, in which endocrinologists play a crucial role. This concept is fundamental not only to optimize patient care, but also to prevent—when possible—the occurrence of significantly invaliding, and sometimes even life-threatening, conditions. In fact, as it emerges from the literature, endocrine irAEs are among the most common ones in oncology, together with cutaneous and gastrointestinal adverse reactions [43]. In the last few years, several papers have summarized the current evidence on the topic, however it is still important to highlight the opportunity for a close endocrine follow-up of patients treated with ICIs [10,27,40,42].

Clinical presentation and management of endocrine irAEs due to ICIs is similar, irrespective of the drug type used, and although several diagnostic and management protocols have been presented through many guidelines for oncologists, there is still no consensus on all the issues mentioned above, due to their peculiar and variable presentation (i.e., routine monitoring for adrenal function in every patient). In fact, their onset is often characterized by non-specific signs and symptoms (i.e., anemia, hyponatremia, fatigue, anorexia, etc.), that can also derive from primary disease and/or can be a consequence of the worsening of cancer patients’ general condition. Moreover, since endocrine irAEs are for the most part chronic conditions, and death is a very uncommon event, they do not quite fit into the established CTCAE 5-grade classification [42,122]. As already proposed, an ideal algorithm for correct patients monitoring should consider the most frequent occurrences, also producing cost-effective results (Figure 2).

In general, baseline assessment of thyroid hormone profiles (TSH, FT4, and TPO-Ab) is recommended before starting ICI therapy, while thyroid function (TSH and FT4) should be checked at each administration. When anti-CTLA-4 is used, thyroid tests should be repeated every 4–6 weeks within 6 months after treatment conclusion, minding late dysfunctions. In case of hypothyroidism, replacement therapy with levothyroxine should be initiated in symptomatic cases (grade > 2), interrupting ICI therapy only with severe symptoms (grade ≥ 3). On the converse, symptomatic hyperthyroidism should be interrupted starting a beta-blocker. A short-term treatment with oral prednisolone 0.5–1 mg/kg may be considered for destructive thyroiditis or with severe symptoms [122]. In addition, blood fasting glucose and electrolyte levels need to be assessed at baseline and during treatment, while evaluation of pituitary function (morning cortisol, ACTH, PRL, gonadotropins, and sex hormones) at baseline is useful for comparison—especially if anti-CTLA-4 is used (hypophysitis can occur within 2–3 months from treatment initiation with anti-CTLA-4, but over 6 months after with anti-PD-1/PD-L1), or in presence of symptoms [10]. With suspected/overt hypophysitis and severe headache, diplopia or neurological symptoms (grade 3), treatment with methylprednisolone 1 mg/kg is indicated. Adrenal crisis from central hypoadrenalism (grade 3) requires stress-dose corticosteroid replacement, while asymptomatic and symptomatic non-severe cases (grade 1–2) replacement of deficient hormones (adrenal, thyroid and gonadal axes) should be started. A similar approach is indicated for primary adrenal insufficiency [122]. However, it would be preferable to regularly check serum electrolytes and morning cortisol levels during treatment, since the most common endocrine dysfunctions (hypophysitis, thyroid dysfunction, and PAI) often cause nonspecific symptoms. Furthermore, primary or secondary adrenal insufficiency should be suspected with hypoglycemia, hyponatremia, fatigue, hypotension, or shock, immediately prompting adequate investigations and corticosteroid administration. In general, hospitalization is not indicated for endocrine irAEs, but it would be advisable with grade ≥ 3 toxicities (i.e., diabetic ketoacidosis or adrenal crisis) for a better management [76]. In milder cases, patients can be followed up at each administration, or more closely if clinically/biochemically required.

## 8. Conclusions

ICI therapy has markedly revolutionized cancer management over the past decade and continues to evolve, including new applications in different tumor types and disease settings. Understanding organ-specific toxicities represents a key goal for both oncologists and many other specialists that can treat this kind of patient during their clinical practice, including endocrinologists. Strict collaboration between specialists is mandatory for the early recognition and proper management of these patients. Despite advances in the understanding of pathogenetic mechanisms underlying endocrine toxicities, these still represent an important challenge for endocrinologists.

Many questions regarding endocrine-related irAEs remain to be unanswered. These include the role of predisposing factors as well as the association between the onset and severity of adverse with effectiveness of ICIs. Also, beyond clinical features, additional tumor predictive biomarkers that can be potentially useful to predict the development of toxicities, are urgently needed. Therefore, well-designed prospective studies should be performed to identify risk factors and biomarkers with a significant predictive value for the incidence of irAEs induced by ICIs, and to determine the implications of the features of the irAEs, in particular with regard to the new drugs, such as anti-LAG-3. This could allow the clinicians to stratify patients according to risk and to take the measures needed to manage the patient experiencing toxicities, in order to avoid the permanent discontinuation of these highly effective therapies in the clinical setting.

## Figures and Tables

**Figure 1 cancers-15-00246-f001:**
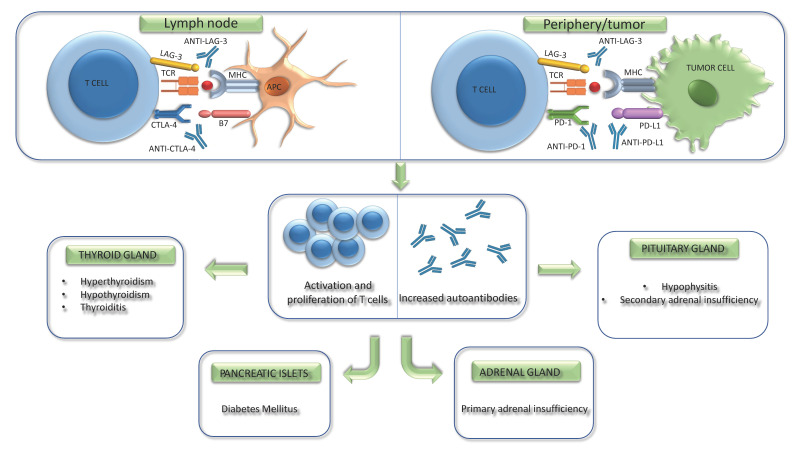
Role of immune checkpoints in antitumor immune responses and endocrine adverse events, induced by immune checkpoint inhibitors. CTLA-4 competes with CD80/CD86 to prevent excessive T-cell activation, while the activation of the PD-1/PD-L1 axis keeps T-cell in an anergic state. Furthermore, LAG-3 interacts with MHC, upregulating the function of T-cells and downregulating TCR signal transduction. Inhibition of these immune checkpoints induced by ICIs prevents inhibition signals, and recognizes T-cells to kill cancer cells. On the other hand, this process can also lead to a reduced self-tolerance against other tissues, thus favoring autoimmune events, including endocrine ones, at different levels. APC: antigen-presenting cell; B7: CD80/CD86; CTLA-4: cytotoxic T-lymphocyte-associated protein; LAG-3: lymphocyte-activation gene 3; MHC: major histocompatibility complex; PD-1: programmed cell death protein 1; PD-L1: programmed death ligand 1; TCR: T cell receptor.

**Figure 2 cancers-15-00246-f002:**
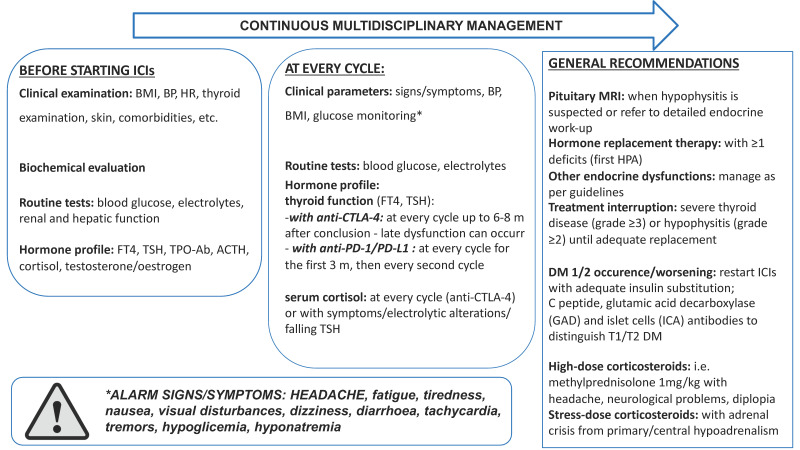
An updated flowchart for endocrine irAEs management, according to the most recent evidence from the literature. BMI: body mass index; BP: blood pressure; HR: heart rate; HPA: hypothalamus-pituitary-adrenal axis; MRI: magnetic resonance imaging; T1/T2 DM: Type1/Type2 diabetes mellitus.

**Table 1 cancers-15-00246-t001:** ICIs approved by FDA.

Target	Agent	Indication	Year of Approval
PD-1	Pembrolizumab	Unresectable or metastatic melanoma (as single agent or in combination with ipilimumab)	2015
		Adjuvant treatment of melanoma (with lymph node(s) involvement) after complete resection	2021
		Advanced Non-Small Cell Lung Cancer (NSCLC)	2016
		Recurrent or metastatic or Head and Neck Squamous Cell Carcinoma (HNSCC)	2016
		Advanced urothelial carcinoma	2017
		Bladder carcinoma non-muscle invasive (high-risk)	2020
		Advanced MSI-H/dMMR cancers	2017
		Advanced MSI-H/dMMR colorectal cancer	2020
		Advanced gastric, esophageal or gastroesophageal junction cancer	2017
		Advanced cervical cancer	2018
		Advanced Hepatocellular Carcinoma (HCC)	2018
		Advanced Merkel cell carcinoma	2018
		Advanced renal cell carcinoma (with axitinib)	2019
		Adjuvant treatment for Renal Cell Carcinoma (RCC)	2021
		Advanced MSI-H/dMMR endometrial carcinoma (with lenvatinib)	2019
		Advanced TMB-H cancers (≥10 mut/Mb)	2020
		Advanced cutaneous Squamous Cell Carcinoma (cSCC)	2020
		Advanced Triple-Negative Breast Cancer (TNBC) or high-risk early-stage TNBC	2020
		Relapsed or refractory classical Hodgkin lymphoma	2017
		Relapsed or refractory primary mediastinal large B-cell lymphoma	2018
	Nivolumab	Advanced melanoma (as single agent or in combination with ipilimumab)	2014
		Advanced NSCLC	2015
		Metastatic RCC after prior therapy	2015
		Relapsed or refractory classical Hodgkin lymphoma	2016
		Advanced or recurrent HNSCC after prior platinum-based therapy	2016
		Locally advanced or metastatic urothelial carcinoma after prior platinum-based chemotherapy	2017
		Advanced MSI-H/dMMR colorectal cancer after prior chemotherapy (as sigle agent or in combination with ipilimumab)	2017
		Advanced HCC after prior treatment with sorafenib (as single agent or in combination with ipilimumab)	2017
		Adjuvant treatment of melanoma [with lymph node(s) involvement] or metastatic disease	2017
		Metastatic or recurrent esophageal squamous cell carcinoma after prior chemotherapy	2020
		Advanced RCC (with cabozantinib)	2021
		Adjuvant treatment of urothelial carcinoma (high risk of recurrence)	2021
		Advanced malignant pleural mesothelioma (with ipilimumab)	2020
		Early-stage NSCLC before surgery	2022
		Unresectable advanced or metastatic esophageal squamous cell carcinoma in combination with chemotherapy or ipilimumab	2022
		Neoadjuvant treatment of resectable NSCLC in combination with platinum-doublet chemotherapy	2022
		Advanced NSCLC (PD-L1 ≥ 1%) or in combination with two cycles of platinum-based chemotherapy	2020
	Cemiplimab	Metastatic or locally advanced cutaneous SCC	2018
		Locally advanced Basal Cell Carcinoma (BCC)	2021
		Metastatic or locally advanced NSCLC (TPS ≥ 50%)	2018
	Dostarlimab	Recurrent or advanced endometrial cancer with mismatch repair deficient (dMMR), after prior platinum-based therapy	2021
PD-L1	Atezolizumab	Metastatic or locally advanced or urothelial carcinoma, after prior platinum-based chemotherapy	2016
		Advanced NSCLC after prior target therapy or platinum-based chemotherapy	2016
		Metastatic or locally advanced or urothelial carcinoma not eligible for platinum- based chemotherapy (expressing PD-L1)	2017
		Advanced NSCLC (squamous) without EGFR or ALK alterations (with carboplatin, paclitaxel and bevacizumab)	2018
		SCLC (extensive disease) (with carboplatin and etoposide)	2019
		Locally advanced or metastatic TNBC (PD-L1 ≥ 1%) (with nab-paclitaxel)	2019
		Advanced NSCLC without EGFR or ALK alterations (with nab-paclitaxel/carboplatin)	2019
		Advanced NSCLC with PD-L1 ≥ 50% and without EGFR or ALK alterations	2020
		Metastatic HCC (with bevacizumab)	2020
		Advanced melanoma BRAF V600 mutation–positive (with cobimetinib and vemurafenib)	2020
		Adjuvant treatment for stage II-IIIA NSCLC (PD-L1 > 1%)	2021
	Durvalumab	Unresectable NSCLC (stage III) after concurrent radiotherapy and platinum-based chemotherapy with nonprogressive disease	2018
		SCLC (extensive disease) (with platinum-based chemotherapy and etoposide)	2020
		Locally advanced or metastatic urothelial carcinoma	2017
		Unresectable HCC (in combination with tremelimumab)	2022
	Avelumab	Metastatic Merkel cell carcinoma	2017
		Locally advanced or metastatic urothelial carcinoma (progression after prior platinum-based chemotherapy or maintenance treatment)	2017
		Advanced RCC (in combination with axitinib)	2019
CTLA-4	Ipilimumab	Unresectable or metastatic melanoma	2011
		Melanoma (stage III) high-risk after complete resection	2015
	Tremelimumab	Unresectable HCC (in combination with durvalumab)	2022
LAG 3	Relatlimab	Unresectable or metastatic melanoma (in combination with nivolumab)	2022

**Table 2 cancers-15-00246-t002:** An overview of the main studies reporting on thyroid irAEs from ICIs treatment.

Year	First Author	Ref.	Patients n.	M/F	ICI	irAEs n. (%)	Thyroid irAEs n. (%)	Hypothyroidism n. (%)	Thyrotoxicosis n. (%)	Previous Thyroid Disease n. (%)
2018	Guaraldi F	[48]	52	22/30	52 (100%) ipilimumab 29 (55.8%) nivolumab for disease progression	-	7 (13.4%)	1 (1.9%, 4 euthyroid HT)) 7 (13.4%)	3 (5.7%) (1 transient)	3 (5.7%)
2019	Yamauchi I	[53]	200	134/66	200 (100%) nivolumab	-	67 (33.5%) 40 (20%) subclinical 27 (13.5%) overt	- - 11 (5.5%) post thyrotoxicosis	- - 17 (8.5%)	NA
2021	Paderi A	[54]	43	35/8	33 (76.7%) nivolumab 10 (23.7%) nivolumab plus ipilimumab	29 (67.4%)	19 (44.2%) endocrine irAEs 15/19 (78.9%) thyroid irAEs	15 (43.88%)	8/19 early thyrotoxicosis	NA
2022	Karhapaa H	[55]	140	75/65	21 (15%) ipilimumab, 46 (33%) nivolumab, 67 (48%) pembrolizumab, and 6 (4%) ipilimumab + nivolumab	-	41 (29.2%) endocrine irAEs 36/41 (87.8%) thyroid irAEs	- 8 (22%)	- 14 (39%)	NA
2021	Ferreira JL	[58]	161		pembrolizumab, nivolumab, and ipilimumab	-	29 (18%)	8.7% primary 4.3% central 2.5% biphasic thyroiditis	2.5%	NA
2021	Luongo C	[59]	96	66/30	67 (69.1%) nivolumab, 18 (18.5%) pembrolizumab, 9 (9.3%) ipilimumab	-	36 (38%)	11 (30.5%)	25 (69.5%) transient	NA
2021	Muir CA	[60]	1246	824/422	165 (13%) ipilimumab, 236 (19%) nivolumab, 448 (36%) pembrolizumab 285 (23%) ipilimumab + nivolumab, and 112 (9%) others	-	518 (42%)	100 (8%)	388 (31%)	NA
2020	Basak EA	[61]	168	103/65	118 (70%) nivolumab 50 (30%) pembrolizumab	-	54 (32%) 34 (20%) subclinical 20 (12%) overt	-	-	27 (16%)
2021	Rubino R	[63]	251	119/62	154 (61.35%) nivolumab 97 (38.65%) pembrolizumab		70 (27.89) endocrine irAEs 66/70 (94.28%) thyroid irAEs	34 (51.52%)	17 (22.72%)	28 (73.68%)

HT: Hashimoto’s thyroiditis; NA: not available.

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
