# Peer review of "Management of Endocrine and Metabolic Toxicities of Immune-Checkpoint Inhibitors: From Clinical Studies to a Real-Life Scenario"

_cancers, 2022, doi:10.3390/cancers15010246_

Round 1

Reviewer 1 Report

The current article is a comprehensive review about the occurrence, frequencies, diagnosis, treatment and prediction of immune related adverse reaction (irAEs) connected to the therapy by ICI (immune check-point inhibitors). Such as, it brings the newest data about all above mentioned aspects of  irAEs. There are included appropriate citations of original source papers with the added value of personal clinical experience with the management of these events. The text has a didactic value and could serve as an algorithm useful for clinical practice.   

No major errors were identified in the text.

Minor comments:

Mention the possibility to administer ICI subcutaneously (r.164-174).

It is necessary to correct some minor inconsistencies, like:

- unify capital and lower-case letters in generic names of drugs:  like ipilimumab/Ipilimumab (should be lower-case) and in names of diseases (Melanoma/melanoma...) – no reason to write with capitals - especially in the tab. 1

- correct disfunction to dysfunction (r. 222), hypoglicemia to hypoglycemia (r.558)

- unify the fonts across the text (different fonts are used, at least I see it in my computer)

Reviewer 2 Report

In this manuscript, the authors describe the endocrine and metabolic toxicities of immune-checkpoint inhibitors.  This is an excellent review of the literature on the potentially long term toxicity of these drugs.  The authors do an exceptionally good job of discussing the toxicities and potential treatments (lines 241-490) and the management of the symptoms (lines 565-621). However, there are some minor changes or additions that would help with comprehension in this overall excellent review.

The sentence on lines 55-58 stating the CTLA-4 functions in the lymph nodes, while PD-1/PD-L1 functions in peripheral tissue is not supported by the references provided or by the substantial data showing expression of CTLA-4 in the periphery and PD-1 in lymph nodes during infections and cancer.  This sentence should be modified or removed from the paper.

The paragraph starting on line 104 appears to ignore a major mechanism for how tumors escape immune surveillance, specifically excluding immune cells.  This is the mechanism, often termed “cold” tumors, used by many tumors to evade the immune response. It is important to mention this mechanism since ICIs are especially ineffective in “cold” tumors.  A short discussion of this phenomenon would be appropriate in this section.

In line 166, the authors state the monoclonal antibodies have a molecular weight of 55 kDa.  This is not true.  A single heavy chain of a monoclonal antibody will have an average molecular weight of 55 kDa, but IgG antibodies are composed of 2 heavy chains and two light chains. The actual average molecular weight of an IgG1, IgG2 and IgG4 antibody is 150 kDa.
